# Nicotine and Cotinine Induce Neutrophil Extracellular Trap Formation—Potential Risk for Impaired Wound Healing in Smokers

**DOI:** 10.3390/antiox11122424

**Published:** 2022-12-08

**Authors:** Romina H. Aspera-Werz, Jonas Mück, Caren Linnemann, Moritz Herbst, Christoph Ihle, Tina Histing, Andreas K. Nussler, Sabrina Ehnert

**Affiliations:** Siegfried-Weller Institute for Trauma Research, BG Trauma Center, University of Tuebingen, Schnarrenbergstrasse 95, 72070 Tuebingen, Germany

**Keywords:** smoking, nicotine, cotinine, neutrophil extracellular traps, reactive oxygen species

## Abstract

Smoking undoubtedly affects human health. Investigating 2318 representative patients at a level 1 trauma center identified delayed wound healing, tissue infections, and/or sepsis as main complications in smokers following trauma and orthopedic surgery. Therefore, smoking cessation is strongly advised to improve the clinical outcome in these patients, although smoking cessation often fails despite nicotine replacement therapy raising the need for specific interventions that may reduce the complication rate. However, the underlying mechanisms are still unknown. In diabetics, delayed wound healing and infections/sepsis are associated with increased neutrophilic *PADI4* expression and formation of neutrophil extracellular traps (NETs). The aim was to investigate if similar mechanisms hold for smokers. Indeed, our results show higher *PADI4* expression in active and heavy smokers than non-smokers, which is associated with an increased complication rate. However, in vitro stimulation of neutrophils with cigarette smoke extract (CSE) only moderately induced NET formation despite accumulation of reactive oxygen species (ROS). Physiological levels of nicotine and its main metabolite cotinine more effectively induced NET formation, although they did not actively induce the formation of ROS, but interfered with the activity of enzymes involved in anti-oxidative defense and NET formation. In summary, we propose increased formation of NETs as possible triggers for delayed wound healing, tissue infections, and/or sepsis in smokers after a major trauma and orthopedic surgery. Smoking cessation might reduce this effect. However, our data show that smoking cessation supported by nicotine replacement therapy should be carefully considered as nicotine and its metabolite cotinine effectively induced NET formation in vitro, even without active formation of ROS.

## 1. Introduction

Smoking undoubtedly affects human health. Besides increasing the risk of certain diseases, e.g., cancers, respiratory, and cardiovascular diseases, smoking also increases the risk of post-surgical complications. In orthopedic and traumatology departments, the number of smokers is higher than that in the common population, suggesting a poor bone quality in smokers [1]. Surgical treatment of a fracture often results in complicated healing, e.g., impaired wound healing, stiffness of joints, and non-unions in smokers [2,3,4]. Consequently, smokers have prolonged hospital stays and require revision surgeries more often than non-smokers.

Reasons for impaired wound healing in smokers have been critically discussed, ever since Mosely and Finseth first reported detrimental effects of smoking on wound healing [5]. The authors claimed a limited blood flow for the observed effects, possibly through the vasoconstrictive action of nicotine [5]. In addition, certain molecules contained in cigarette smoke, in particular nicotine, carbon monoxide, and hydrogen cyanide, have been reported to induce ischemia in the tissues by inhibiting proliferation of red blood cells and limiting oxygen transport and metabolism at the cellular level [6] and to exert direct toxic effects on the cells involved in the healing process, e.g., fibroblasts, keratinocytes, endothelial cells, and, additionally, immune cells [7].

Focusing on mechanisms of impaired wound healing, diabetic patients represent a good reference. In diabetics, there is a clear link between impaired wound healing and excessive formation of so-called neutrophil extracellular traps (NETs) by neutrophils infiltrating the wound tissue—for review, see [8]. NETs are comprised of DNA covered with citrullinated histones and anti-microbial peptides, released from neutrophils with increased peptidyl-arginine-deiminase 4 (*PADI4*) expression, calcium influx, and/or oxidative stress. Their physiological role is to rapidly and unspecifically bind and neutralize pathogens [9]. Smoking has been reported to strongly affect the function of neutrophils, e.g., chemotaxis, phagocytosis, and inflammatory response [10]. Increased NET formation is described in smokers with chronic obstructive pulmonary disease, thrombosis, or atherosclerosis [11,12], but has not yet been associated with impaired wound healing.

Smokers are strongly advised to quit smoking to not hinder the healing process following a surgery. A recent meta-analysis investigated the effect of eight different pre-admission interventions on the clinical outcomes after elective surgeries [13]. The analysis clearly showed that smoking cessation reduced wound infections and delayed wound healing by 72% on average (CI: 36–88%). However, one needs to consider that smoking is a powerful addictive drug, with nicotine being the addictive molecule in cigarette smoke. Therefore, the attempt to quit smoking frequently fails. Only 8% of smokers were able to successfully quit smoking [14]. Accompanying nicotine-replacement therapy, e.g., with nicotine transdermal patches, sprays, gums, electronic cigarettes or tobacco heating devices might double the success rate for smoking cessation [15].

It is indisputable that nicotine replacement therapy may support smoking cessation; however, nicotine itself may act as a cellular toxin and thus interfere with the healing process in patients. Therefore, this study aimed to investigate effects of smoking on wound healing in 2318 randomly assigned patients at a level 1 trauma center. As possible regulatory mechanisms, smoking-dependent effects on neutrophilic *PADI4* expression will be investigated. In vitro, the effects of cigarette smoke extract (CSE), nicotine, and cotinine on formation of NETs and associated oxidative stress response will be investigated.

## 2. Materials and Methods

If not indicated differently, chemicals were obtained from Merck (Darmstadt, Germany).

### 2.1. Patient Data and Human Material

Complication rates were initially determined from a retrospective analysis of data obtained from the hospital information system. The patients’ characteristics are summarized in Table 1. Smoking was recorded as packyears (1 PY equals 20 cigarettes per day for 1 year or a total of 7305 cigarettes).

For analyzing the role of neutrophils in this process, EDTA venous blood was prospectively collected from patients and healthy volunteers after informed consent. As diabetes mellitus is known to increase *PADI4* expression, the number of diabetics was kept at a minimum (only included when good blood glucose control/number equalized between the groups). The donor matching and patient characteristics for the in vitro experiments are summarized in Figure 1. All experiments were carried out with the permission of the Ethics Committee of the University Hospital of Tübingen (Ethics vote: 666/2018B02).

### 2.2. Isolation of Neutrophils

Neutrophils were isolated from fresh EDTA venous blood by density gradient centrifugation [16]. Briefly, the EDTA venous blood was carefully layered on Lympholyte Poly isolation medium (#CED-CL5075, Cedarlane, ON, Canada) in a 1:1 ratio. Samples were centrifuged for 35 min at 500× *g* without break at ambient temperature. The plasma and peripheral blood mononuclear cell layers were discarded. The polymorphonuclear cell (PMN) layer was carefully transferred to a 15 mL tube and washed twice with PBS (12 mL, centrifugation at 450× *g*, 10 min, ambient temperature without break). Isolated cells were resuspended in RPMI-1640 medium without phenol red (#R8758) containing 2% autologous plasma and quantified by Trypan Blue exclusion method. The isolated neutrophils were diluted to obtain a density of 1 × 10^6^ cells/mL (concentrated cell suspension).

### 2.3. PADI4 Gene Expression

Phenol–chloroform extraction method was used to isolate total mRNA from the freshly isolated neutrophils. After photometric quantification, 500 ng of the total mRNA was converted into cDNA using the first-strand cDNA synthesis kit (#K1612, ThermoFisher Scientific, Karlsruhe, Germany). Quantitative RT-PCRs were performed in the StepOnePlus^TM^ qPCR cycler using the Green Master Mix (2X) High Rox (#M3052.0500, Genaxxon, Ulm, Germany). PCR conditions are summarized in Table 2.

The specificity of the qPCR reactions was checked by melting curve analyses. Relative *PADI4* expression levels were calculated using the 2^−ΔΔCt^ method, using *EF1α* and *RPL13a* as housekeeping genes, as these proved to be most stable (GeNorm, NormFinder, Best Keeper, comparative Δ_Ct_ method [17]) between individuals (Figure 2).

### 2.4. Stimulation of Neutrophils

CSE was prepared as described before [18]. Briefly, the smoke of 2 commercial cigarettes (Marlboro red, Philip Morris New York City, NY, USA) was bubbled (100 bubbles/min) through 50 mL of plain RPMI-1640 medium using a wash bottle. A CSE solution with an optical density of λ_320nm_ = 0.7 was considered as 100% CSE. After being sterile filtered, the CSE was mixed with RPMI-1640 culture medium to obtain the stimulation solution, of which 10% CSE was considered as smoking 20 cigarettes per day. As a positive control for NET formation, neutrophils were stimulated with 100 nM PMA (Phorbol 12-myristate 13-acetate, #Cay10008014, Biomol, Hamburg, Germany). All solutions were prepared as 1.25-fold of the final concentration to allow measurement of blank (80 µL) and thereafter addition of concentrated cell suspension (20 µL) to receive a final concentration of 2 × 10^5^ neutrophils/mL [16].

### 2.5. SYTOX^TM^ Green Assay

For measuring DNA release, neutrophils (2 × 10^5^ cells/mL) were incubated with SYTOX^TM^ Green at a final concentration of 1 µM. For normalization, neutrophils were lysed with 1% Triton-X-100 (Carl Roth, Karlsruhe, Germany). Fluorescence (λ_ex_ = 485 nm and λ_em_ = 520 nm) was measured every 30 min with the Omega Plate Reader (BMG Labtech, Ortenberg, Germany) at 37 °C and 5% CO_2_. Curve fitting of the resulting XY diagrams with GraphPad Prism included the maximum NET release rate (maximum slope) and the total DNA released (area under the curve—AUC) [16].

### 2.6. Immuno-Fluorescent Staining

Neutrophils (3 × 10^5^ cells/mL) were seeded on poly-L-lysine coated chamber slides (150 µL/well) and stimulated as indicated. After incubation at 5% CO_2_, 37 °C for 5 h, the cells were fixed with 4% formaldehyde solution for 30 min at ambient temperature. Then, cells were permeabilized with 0.5% Triton-X-100 solution for 5 min. Next, 5% BSA was used for blocking (1 h) before incubation with anti-myeloperoxidase antibody (dilution 1:200 in PBS, sc-52707, Santa Cruz Biotechnology, Heidelberg, Germany) overnight at 4 °C. After being washed 3 times with PBS, cells were incubated for 2 h with AlexaFluor-488 anti-mouse antibody (1:1000, #A-10667, ThermoFisher Scientific) and Hoechst 33342 (2 µg/mL, #14533) in PBS. After a final washing step, fluorescent images were captured with EvosFL fluorescent microscope.

### 2.7. DCFH-DA Assay

Formation of reactive oxygen species (ROS) were quantified with the DCFH-DA assay. Briefly, neutrophils were loaded with 5 μM 2′,7′-dichlorofluorescein-diacetate (DCFH-DA, #D6883) in RPMI-1640 Medium for 20 min at 37 °C; 5% CO_2_. To remove excess DCFH-DA, the cells were washed once with PBS (400 g, 10 min, 50% breaks). Then, the cells were distributed on a 96-well plate containing the stimuli, and fluorescence (λ_ex_ = 485 nm and λ_em_ = 520 nm) was measured immediately (1 min intervals for 20 min) in the Omega Plate Reader at 37 °C.

### 2.8. Enzyme Activity Assays

Activities of isolated/synthetic enzymes involved in the anti-oxidative defense or NET formation were measured in the following assays, in presence or absence of 20% CSE, 200 ng/mL nicotine (#8746.1, Carl Roth), or 200 ng/mL cotinine (#L11873.0, VWR, Radnor, PE, USA).

#### 2.8.1. Superoxide Dismutase (SOD) Activity Assay

SOD activity was determined using a commercial test kit (#19160-1KT-F, Merck). The increase in absorbance (λ = 450 nm) was detected with the Omega Plate Reader [19].

#### 2.8.2. Catalase Activity Assay

Catalase activity was measured with the OxiSelect^TM^ Catalase test kit (#STA-339, Cell Biolabs, San Diego, CA, USA). The increase in fluorescence (λ_ex_ = 544 nm and λ_em_ = 590 nm) was quantified with the Omega Plate Reader [19].

#### 2.8.3. Glutathione Peroxidase (GPx) Activity Assay

Briefly, 5 µL of the GPx solution was mixed with 5 µL of the CSE, nicotine, or cotinine solution. Then, 15 μL of a 4 mM NADPH solution and 75 μL of the GPx assay solution (1.3 U/mL glutathione reductase, 1.3 mM reduced L-glutathione in 0.05 mM potassium phosphate buffer (pH = 7.0) containing 1.1 mM EDTA and 1.1 mM NaN_3_) were added. The mixture was incubated at ambient temperature for 5 min before the addition of 10 μL of the 15 mM cumene hydroperoxide solution (#247502, Merck). The decrease in absorbance (λ = 340 nm) was measured with the Omega Plate Reader [20,21].

#### 2.8.4. Glutathione Reductase (GR) Activity Assay

Briefly, 2.5 μL of the GR solution was mixed with 2.5 μL of the CSE, nicotine, or cotinine solution. Then, 185 μL of the reaction mixture (0.8 mM 5′-dithiobis-(2-nitrobenzoic acid)—DTNB, #6334.1, Carl Roth), 0.1 mM NADPH, and 1 M EDTA in 0.2 M potassium phosphate buffer (pH = 7.5) and 10 μL of a 20 mM reduced L-glutathione solution were added. The following increase in absorbance (λ = 412 nm) was detected with the Omega Plate Reader [21,22].

#### 2.8.5. Peptidyl-Arginine-Deiminase 4 (PADI4) Activity Assay

Briefly, 15 μL of the neutrophil lysate was mixed with 15 μL of a CSE, nicotine, or cotinine solution in a black microwell plate. After the addition of 50 μL of the reaction mixture I (100 mM NaCl, 100 mM TRIS, 20 mM CaCl_2_, and 1 mM Tris(2-carboxyethyl)phosphine hydrochloride—TCEP, #C4706 (pH = 8.0)), the mixture was incubated at ambient temperature for 20 min. Then, 20 μL of reaction mixture II (250 µM Z-Arg-Arg-7-amido-4-methylcoumarin hydrochloride—ZRCoum, #C5429 in ddH_2_O) was added. After 45 min incubation in the dark, final 5 µL of a trypsin solution (10 mg/mL bovine trypsin in 100 mM EDTA) was added to each well. After another 20 min of incubation in the dark, fluorescence (λ_ex_ = 355 nm and λ_em_ = 460 nm) was measured with the Omega Plate Reader [23].

#### 2.8.6. Myeloperoxidase (MPO) Activity Assay

MPO activity was measured with the Myeloperoxidase Activity Assay Kit II (#PK-CA577-K745, PromoCell, Heidelberg, Germany) using 15 µL of neutrophil lysate. The change in fluorescence (λ_ex_ = 485 nm and λ_em_ = 520 nm) over 40 min was quantified with the Omega Plate Reader.

#### 2.8.7. Neutrophilic Elastase (ELANE) Activity Assay

Briefly, 25 μL of the neutrophil lysate was mixed with 25 μL of a CSE, nicotine, or cotinine solution in a black microwell-plate. Then, 50 μL of the reaction solution (0.4 mM MeOSuc-AAPV-AMC (#sc-201163, Santa Cruz Biotechnology), 0.1 M TRIS buffer (pH = 7.4)) were added. The change in fluorescence (λ_ex_ = 355 nm and λ_em_ = 460 nm) over 1 h was quantified with the Omega Plate Reader.

### 2.9. Statistical Analysis

The numbers of patients/donors (*N*) and technical replicates (*n*) for each experiment are given in the figure legends. Patient data are displayed in violin plots and in vitro data as box plots with individual measurement points (average for each *N*)—each with median and interquartile range. As Gaussian distribution was not provided, data were compared by non-parametric Kruskal–Wallis test with Bonferroni correction for multiple comparisons. Complication rates in the different groups were compared by chi-squared (*χ*^2^) test. All statistical analyses were performed with GraphPad Prism Version 8 (San Diego, CA, USA). A *p* < 0.05 was considered significant.

## 3. Results

### 3.1. Smoking Increases the Complication Rates in Orthopedic and Trauma Patients

To obtain an overview on the current situation of smokers at a level 1 trauma center, data from 2318 randomly assigned patients were analyzed. Regarding the smoking behavior, 51.3% of the patients were non-smokers (0 PY), 29.5% of the patients were moderate smokers (<20 PY), and 18.8% of the patients were heavy smokers (≥20 PY/Figure 3A). In these patients, the development of complications was analyzed in groups, i.e., delayed/impaired bone or wound healing, infections/sepsis, implant-associated complications, and other complications. Overall, the complication rate was significantly higher in heavy smokers than in moderate smokers or non-smokers. The most frequent complications were associated with delayed/impaired wound healing with 12.1% of non-smokers, 16.1% of moderate smokers, and 23.0% of heavy smokers affected. By numbers, the second most frequent complications involved tissue infections and sepsis, with 8.8% of non-smokers, 10.2% of moderate smokers, and 20.7% of heavy smokers affected (Figure 3B).

### 3.2. Patient Characteristics

Impaired wound healing in diabetics is associated with excessive NET formation [8], induced by over-expression of neutrophilic *PADI4*. In order to investigate if a similar mechanism is induced by smoking, patients were prospectively recruited to analyze the neutrophils prior to surgery. Overall, 117 patients were screened, but from 11 patients, no blood sample was obtained. From the remaining 106 patients, another 16 had to be excluded due to several reasons, e.g., an acute infection (*N* = 7), cancer (*N* = 5) or another NET-associated disease (*N* = 3). The remaining 90 patients were paired into non-smokers and smokers with best possible match of age, gender, as well as number of comorbidities and daily drug consumption (Figure 1A). In line with the large screening cohort, the number of complications was higher in smokers (26.2%) than non-smokers (7.1%/Figure 1B,C).

### 3.3. Neutrophilic PADI4 Expression Is Increased in Heavy and Active Smokers

Neutrophils were isolated from fresh venous EDTA blood obtained from the patients prior to surgery. Neutrophilic *PADI4* expression was determined by qRT-PCR with *EF1α* and *RPL13a* as housekeeping genes. *PADI4* expression was significantly higher (2.4-fold) in heavy smokers than in non-smokers (Figure 4A). Moderate smokers, which contained several former smokers, had a *PADI4* expression comparable to the controls. Consequently, *PADI4* expression was significantly higher in active smokers (2.2-fold) than in non-smokers or former smokers (Figure 4B). In line with the observation in diabetics, *PADI4* expression was higher in patients that developed a complication than in patients that did not develop a complication (Figure 4C). In addition, the effects of age, body mass index (BMI), alcohol consumption, or number of comorbidities and daily drugs on *PADI4* expression were analyzed. Age or alcohol consumption had no effect on *PADI4* expression. While *PADI4* expression tendentially decreased with increasing BMI, the opposite trend was observed with the number of comorbidities and daily drugs (Figure 4D).

### 3.4. CSE, Nicotine, and Cotinine Moderately Induce NET Formation

Freshly isolated neutrophils were stimulated with different physiological concentrations of CSE, nicotine, and its metabolite cotinine. NET formation was quantified using the SYTOX^TM^ Green assay. The assay revealed that all three substances dose-dependently induced DNA release; however, the amount of DNA released was much lower than that observed in the positive control stimulated with PMA. Interestingly, the DNA release induced by nicotine and cotinine was approx. double the amount of the DNA release induced by the respective amount of CSE (Figure 5A–C). This finding was confirmed by immuno-fluorescent stainings for DNA and MPO (myeloperoxidase), where neutrophils stimulated with 200 ng/mL nicotine or 200 ng/mL cotinine formed more NET-like structured than neutrophils stimulated with 20% CSE (Figure 5D).

### 3.5. CSE Induces Oxidative Stress in Neutrophils and Interferes with the Cells Antioxidative Defense

Smoking is well known to induce oxidative stress, a cofactor for NET formation. As expected, exposure to CSE dose-dependently induced ROS in neutrophils. As observed before, the effect was less pronounced than with PMA stimulation (Figure 6A). In contrast, stimulation with nicotine and cotinine dose-dependently reduced ROS in neutrophils (Figure 6B,C). The molecules contained in CSE might interfere with the activities of enzymes involved in the cells anti-oxidative defense. The activity of superoxide dismutase was not significantly affected by 20% CSE, 200 ng/mL nicotine, or 200 ng/mL cotinine (Figure 6D). However, all three substances inhibited the activity of catalase, with the strongest effect observed for CSE (Figure 6E). Similarly, all three substances inhibited the activity of glutathione reductase. However, here, the strongest effect was observed for cotinine (Figure 6G). Glutathione peroxidase activity was not affected by nicotine or cotinine, but was strongly induced by CSE (Figure 6F).

### 3.6. CSE, Nicotine and Cotinine Affect the Activity of Enzymes Involved in NET Formation

As CSE, nicotine, and cotinine inhibited the activity of several anti-oxidative enzymes, their effect on the activity of enzymes involved in NET formation was measured. PADI4 activity was enhanced significantly by CSE (+42%), and tendentially by nicotine (+42%) and cotinine (+25%/Figure 7A). While MPO activity seemed to be inhibited by CSE (−22%), it was tendentially enhanced by nicotine (+57%) and cotinine (+20%/Figure 7B). ELANE activity was inhibited by CSE (−33%), nicotine (−31%), or cotinine (−41%/Figure 7C).

## 4. Discussion

Our data clearly show that smokers have an increased rate of complications, above all delayed wound healing and/or infections/sepsis. This demonstrates that pathogen defense mechanisms are altered in smokers, since neutrophil levels have been reported to be higher in smokers compared to non-smokers [24]. A recent meta-analysis clearly showed that smoking cessation may reduce the occurrence of wound infections and wound healing disorders by up to 88% [13], therefore smokers are strongly advised to stop smoking when facing a large surgery. However, this attempt frequently fails. It is even estimated that smokers need more than 30 quit attempts on average before being successful [25]. In fact, another literature review even suggested that it may be harder to quit smoking than to stop using cocaine or opiates—success rates for opiates/cocaine or alcohol cessation were >44% and 18%, respectively; however, only 8% of smokers were able to successfully quit smoking [14]. Nicotine is the addictive molecule in cigarette smoke. The nicotine contained in tobacco smoke is easily absorbed into and distributed with the blood flow, eliciting pleasant feelings in the smoker within seconds after taking a puff. However, nicotine is quickly metabolized to cotinine and its effects start to subside within a few minutes. This contributes to the strong physical and emotional withdrawal symptoms smokers experience when attempting to quit. The almost unrestricted access to the drug further aggravates the quitting process. Therefore, patients might compensate for the nicotine with nicotine replacement products, e.g., nicotine transdermal patches, sprays, gums, electronic cigarettes or tobacco heating devices. Although all replacement products provide nicotine, big differences in the success rate for smoking cessation have been reported. Overall, nicotine gums, but even more so electronic cigarettes and tobacco heating devices seem to be superior to the classical nicotine sprays or transdermal patches supporting smoking cessation [26].

In our patients, basal (prior to surgery) neutrophilic *PADI4* expression was increased in patients who smoked, especially when they have been heavy (≥20 PY) and active smokers. The highest neutrophilic *PADI4* expression levels were observed in those patients that developed a respective complication. In diabetics, increased neutrophilic *PADI4* expression and related excessive NET formation have been linked to delayed wound healing and/or infections/sepsis [8]. In smokers, such a link between smoking and *PADI4* expression or activity cannot be easily assumed. While an increased *PADI4* expression in bronchoalveolar lavage fluid in smokers with rheumatoid arthritis-associated interstitial lung disease argues for it, the missing link to the citrullinated auto-antibodies confirming PADI4 activity in these patients argues against it [27]. One possible explanation for this contradictory observation might be the increased levels of anti-PADI4 antibodies found in blood of smokers with rheumatoid arthritis [28].

In contrast to rheumatoid arthritis, the readout for PADI4 activity during wound healing is not the formation of citrullinated auto-antibodies but the formation of NETs. In vivo, increased NET formation was shown to contribute to thrombosis [29] and chronic pulmonary inflammation in smokers with chronic obstructive pulmonary disease [12]. In our setup, stimulation of isolated neutrophils with CSE only slightly (approx. 1.2-fold) induced NET formation when compared to classical NET-stimuli, e.g., PMA (4.3-fold). This was surprising, as CSE effectively induced the formation of ROS, which are required for and may independently trigger NET formation [30]. In the lungs, cells are more or less directly exposed to cigarette smoke, which is assumed to contain more than 10^17^ oxidant and toxic molecules per puff [31]. Therefore, it is not surprising that cells in the lung are massively affected by cigarette smoke. However, when absorbed in the lungs and entering the blood flow, these components are rapidly metabolized. For example, 70–80% of the absorbed nicotine is converted to its main metabolite cotinine when passing through the liver. The half-life of resulting cotinine (~16 h) is approx. eight times longer than that of nicotine. Therefore, smokers’ blood usually contains less nicotine (50–100 ng/mL) than cotinine (250–300 ng/mL) [32]. Several studies demonstrated an association between cigarette smoke and oxidative stress-induced tissue damage, which is supported by our data showing increased ROS formation in neutrophils exposed to CSE.

Interestingly, nicotine and cotinine alone did not actively form ROS, but more effectively induced NET formation in neutrophils by 2.2-fold and 1.9-fold, respectively. However, the NET formation rate has still to be considered moderate. In the literature, it is assumed that nicotine effectively induces NET formation in a ROS-dependent manner [33]. Both nicotine and cotinine have been shown to indirectly contribute to the accumulation of ROS in CSE-exposed mesenchymal stromal cells by inhibiting the activity of certain anti-oxidative enzymes [34]. In line with this observation, physiological levels of nicotine and cotinine did not actively induce ROS in neutrophils but inhibited the activity of catalase and glutathione reductase. The SOD activity was not significantly affected by CSE, nicotine, or cotinine, which allows conversion of •O_2_^−^ to H_2_O_2_, which is normally further processed to H_2_O directly by catalase or under consumption of glutathione by GPx. As both pathways are either directly (inhibition of catalase) or indirectly (inhibition of the recycling of glutathione disulfide) affected by nicotine and cotinine, this might lead to an accumulation of H_2_O_2_ in the neutrophils. As ROS might independently induce NET formation [30], the observed accumulation in H_2_O_2_ may explain the moderate NET formation in neutrophils exposed to physiological concentrations of CSE, nicotine, or cotinine. One factor that might contribute to the unexpectedly low NET formation rate is the extracellular pH; lowering the pH, as commonly observed in smokers, was able to experimentally reduce ROS-dependent NET formation [35].

The fact that nicotine and cotinine inhibit the activity of anti-oxidative enzymes and the expression they induce suggests the possibility that these substances might also influence the activity of enzymes involved in NET formation, e.g., PADI4, ELANE, or MPO. While no reports on the effect of nicotine and cotinine on PADI4 activity could be found, our data suggest that PADI4 activity might be enhanced in smokers. In our setting presence of the full CSE inhibited MPO activity, however, nicotine and cotinine alone seemed to enhance MPO activity. This is in line with a report showing that nicotine significantly affected MPO activity, contributing to sepsis-induced oxidative multiorgan damage after acute administration. Interestingly, in this study, chronic administration to lower doses of nicotine or its withdrawal had the adverse effect [36]. This suggests that smoking itself might alter the responsiveness of the immune cells, which represents a major limitation of our study, where in vitro experiments only simulated smoking conditions on neutrophils isolated from non-smokers. In line with the aforementioned study, acute administration of nicotine in a mouse model reduced MPO activity and protected from lipopolysaccharide-induced acute lung injury [37]. These reports might favor smoking cessation with the help of nicotine replacement products which can increase the success rate for smoking cessation [15]; however, the nicotine release kinetics might be crucial. While nicotine sprays show nicotine plasma kinetics similar to cigarettes, the nicotine plasma levels obtained from nicotine gums or patches are less high but prolonged [38]. Furthermore, the results have to be observed with care, as mice and rats can cope with much higher nicotine levels (EC_50_: 3.3 or 50 mg/kg, respectively) than humans (EC_50_: 0.8 mg/kg) [39]. ELANE expression and activity were reported to be increased by nicotine in vivo [40]; however, the opposite was observed in our setting. Presence of physiological concentrations of CSE, nicotine, or cotinine effectively inhibited ELANE activity, which could partly explain the only moderate NET formation observed with these stimuli.

## 5. Conclusions

In summary, our data show high complication rates in smokers after orthopedic and trauma surgery, with wound healing disorders, tissue infections, and/or sepsis as the most common complications. A possible cause might be increased formation of NETs, triggered by increased ROS formation and interference with anti-oxidative defense mechanisms. Smoking cessation might reduce this effect. However, our data show that smoking cessation supported by nicotine replacement therapy should be carefully deliberate as nicotine and its metabolite cotinine induced NET formation in vitro, even without active formation of ROS.

## Figures and Tables

**Figure 1 antioxidants-11-02424-f001:**
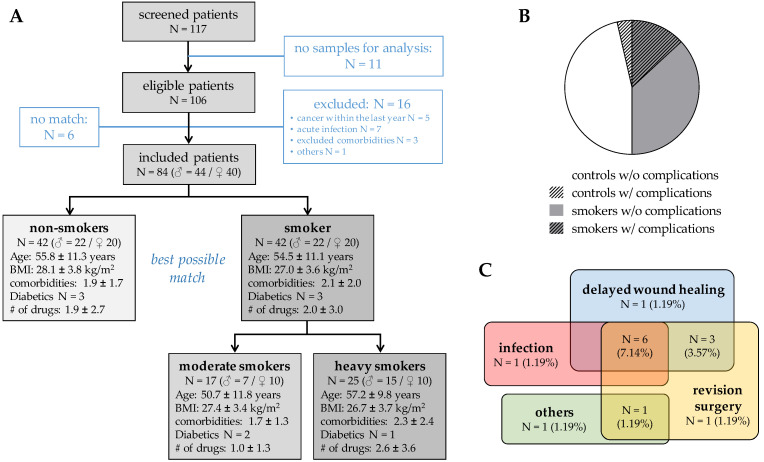
Overview of the patients prospectively collected to analyze the effect of smoking on neutrophilic *PADI4* expression. (**A**) CONSORT diagram on the patients analyzed in this study. Smoking was recorded as packyears (PY). The complications rates in non-smokers and smokers are summarized as (**B**) pie chart and additionally as (**C**) VENN diagram to show multiple complications in one individual.

**Figure 2 antioxidants-11-02424-f002:**
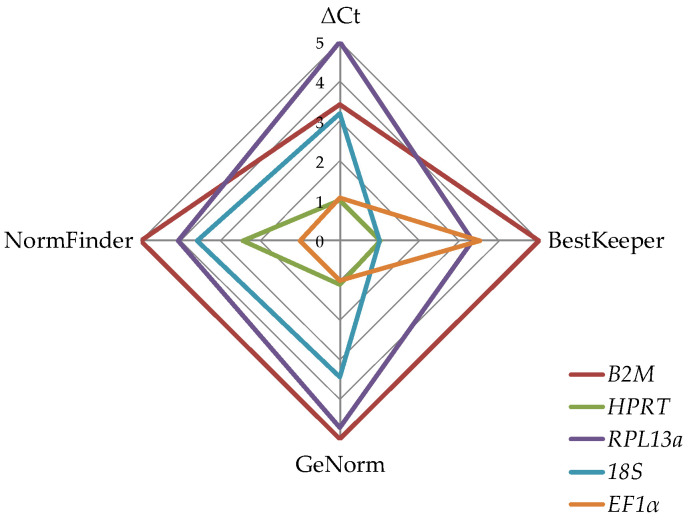
Inter-individual stability of the housekeeper genes. Five different housekeeping genes (*B2M*, *HPRT*, *RPL13*, *18S*, and *EF1α*) were tested with cDNA from 10 different donors. The stability of the housekeeper genes was analyzed by ΔCt, BestKeeper, GENorm, and NormFinder methods. The results are presented as spider-web diagram with the stability values as axis. In summary, the housekeeping genes with the best inter-individual stability were *RPL13* and *EF1α*.

**Figure 3 antioxidants-11-02424-f003:**
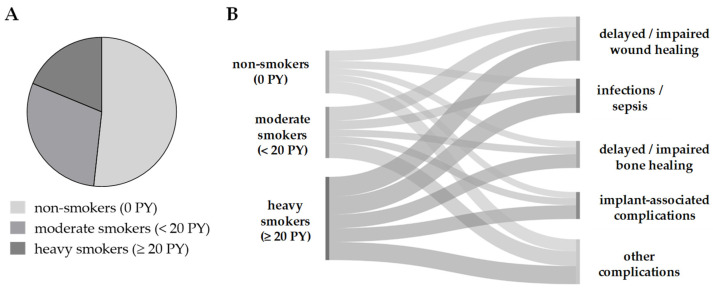
Effect of smoking on the complication rate in patients following orthopedic and trauma surgery. Data on smoking behavior and post-surgical complications from 2318 representative patients at a level 1 trauma center have been obtained from the hospital information system. (**A**) Smoking was recorded as packyears (PY): 51.3% of the patients claimed to be non-smokers (0 PY), 29.5% of the patients were considered as moderate smokers (<20 PY), and 18.8% of the patients were considered as heavy smokers (≥20 PY). (**B**) Complications were summarized as groups and are presented in a cord diagram (made with SankeyMATIC, https://sankeymatic.com/build/, accessed on 20 October 2022), where the cord width equals the complication rate in % within the individual group (non-, moderate, or heavy smokers).

**Figure 4 antioxidants-11-02424-f004:**
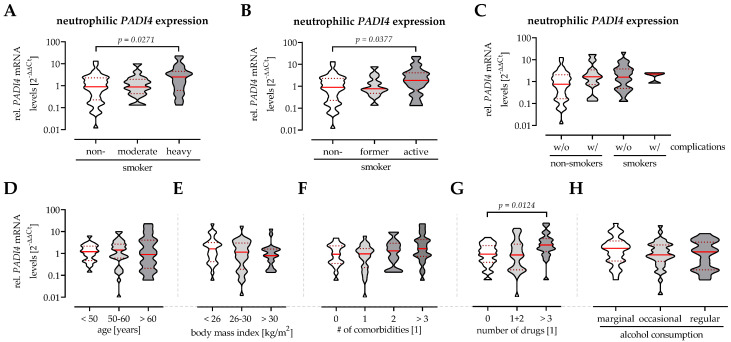
Influencing factors on neutrophilic *PADI4* expression in orthopedic and trauma patients. Neutrophilic *PADI4* expression was analyzed by qRT-PCR using the 2^−ΔΔCt^ method with *EF1α* and *RPL13a* as housekeeping genes and non-smokers as reference. To investigate the effect of smoking, samples were divided according to the patients smoking behavior, differentiating (**A**) non-smokers (0 PY), moderate smokers (<20 PY), and heavy smokers (≥20 PY); (**B**) non-smokers, former smokers and active smokers, and (**C**) non-smokers and smokers that did or did not develop complications. To identify other factors that possibly affect neutrophilic *PADI4* expression, samples were additionally grouped by (**D**) age, (**E**) body mass index, number of (**F**) comorbidities and (**G**) daily drugs, and (**H**) alcohol consumption. Data are presented as violin blots with median and interquartile range. Groups were compared by non-parametric Kruskal–Wallis test followed by Bonferroni multiple comparison test. A *p* < 0.05 was considered as significant as is presented in the graphs.

**Figure 5 antioxidants-11-02424-f005:**
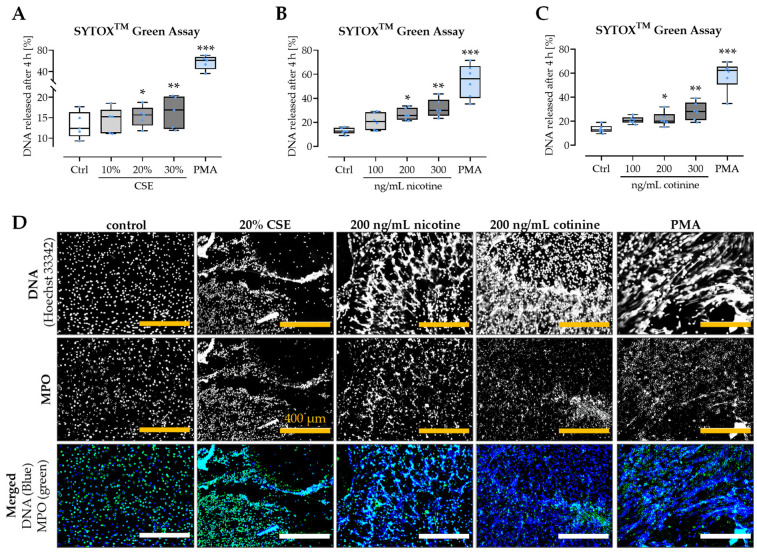
Induction of NET formation by CSE, nicotine, and cotinine. Neutrophils were isolated from healthy volunteers and stimulated for NET formation with freshly prepared (**A**) CSE (10, 20, and 30%), (**B**) nicotine (100, 200, and 300 ng/mL), or (**C**) cotinine (100, 200, and 300 ng/mL). As a positive control, cells were stimulated with 100 nM PMA. NET formation was quantified with the SYTOX^TM^ Green assay (*N* ≥ 5, *n* = 6). Data are presented as box blots with individual measurement points, median, and interquartile range. Groups were compared by non-parametric Kruskal–Wallis test followed by Bonferroni multiple comparison test. * *p* < 0.05, ** *p* < 0.01, and *** *p* < 0.001 as compared to the respective control group (Ctrl). NET formation was additionally confirmed by immuno-fluorescent imaging for DNA (Hoechst 33342—blue) and MPO (green). Representative images are given in (**D**). Scale bar = 400 µm.

**Figure 6 antioxidants-11-02424-f006:**
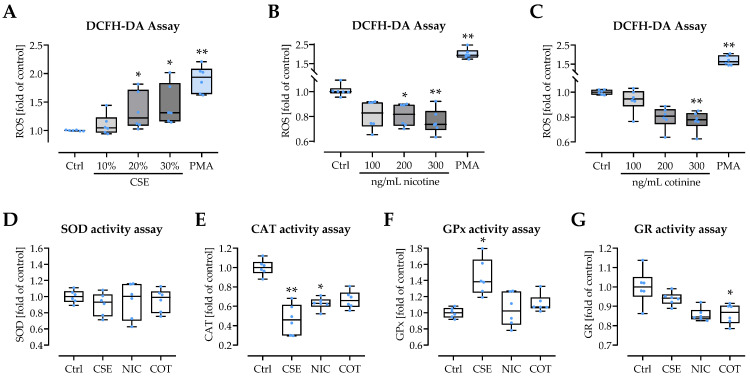
Influence of CSE, nicotine, and cotinine on neutrophilic ROS formation and anti-oxidative defense mechanisms. Neutrophils were isolated from healthy volunteers, loaded with DCFH-DA, and then stimulated with freshly prepared (**A**) CSE (10, 20, and 30%), (**B**) nicotine (100, 200, and 300 ng/mL), or (**C**) cotinine (100, 200, and 300 ng/mL) to quantify formed ROS. The effects of 20% CSE, 200 ng/mL nicotine (NIC), or 200 ng/mL cotinine (COT) on the activity of isolated (synthetic) enzymes of the anti-oxidative defense were determined. Enzyme activities analyzed were (**D**) superoxide-dismutase (SOD), (**E**) catalase (CAT), (**F**) glutathione peroxidase (GPx), and (**G**) glutathione reductase (GR). (*N* = 6, *n* = 6). Data are presented as box blots with individual measurement points, median, and interquartile range. Groups were compared by non-parametric Kruskal–Wallis test followed by Bonferroni multiple comparison test. * *p* < 0.05 and ** *p* < 0.01 as compared to the respective control group (Ctrl).

**Figure 7 antioxidants-11-02424-f007:**
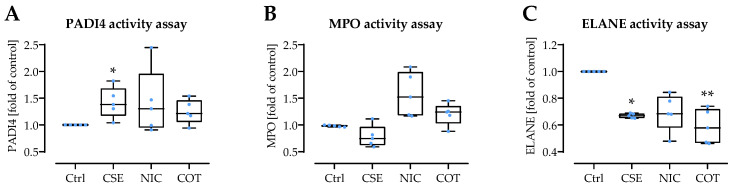
Influence of CSE, nicotine, and cotinine on activity of enzymes involved in NET formation. The effects of 20% CSE, 200 ng/mL nicotine (NIC), or 200 ng/mL cotinine (COT) on the activity of isolated enzymes involved in NET formation were determined. Enzyme activities analyzed were (**A**) peptidyl-arginine-deiminase 4 (PADI4), (**B**) myeloperoxidase (MPO), (**C**) neutrophilic elastase (ELANE). (*N* = 4, *n* = 2). Data are presented as box blots with individual measurement points, median, and interquartile range. Groups were compared by non-parametric Kruskal–Wallis test followed by Bonferroni multiple comparison test. * *p* < 0.05 and ** *p* < 0.01 as compared to the respective control group (Ctrl).

**Table 1 antioxidants-11-02424-t001:** Characterization of the patients included in the retrospective data analysis from the hospital information system. Data are presented as absolute numbers [*N*] and percentage or as mean ± standard deviation (range). For complications, one patient might have multiple codings.

	Non-Smokers	Moderate Smokers	Heavy Smokers	All
Number of patients [*N*]	1199	684	435	2318
Male [*N*]	561 (46.8%)	443 (64.8%)	304 (69.9%)	1308 (56.4%)
Female [*N*]	638 (53.2%)	241 (35.2%)	131 (30.1%)	1010 (43.6%)
Age [years]	59.0 ± 18.6 (18–96)	53.0 ± 17.8 (18–97)	58.0 ± 11.9 (24–91)	57.0 ± 17.5 (18–97)
Body mass index [kg/m^2^]	27.3 ± 5.3 (16.0–52.3)	27.6 ± 5.4 (15.6–60.8)	28.2 ± 5.7 (14.0–48.3)	27.6 ± 5.4 (14.0–60.8)
Diabetes mellitus [*N*]	149 (12.4%)	65 (9.5%)	88 (20.2%)	302 (13.0%)
Complications with:				
Wound healing [*N*]	145 (12.1%)	110 (16.1%)	100 (23.0%)	355 (15.3%)
Infections/Sepsis [*N*]	106 (8.8%)	70 (10.2%)	90 (20.7%)	266 (11.5%)
Bone healing [*N*]	87 (7.3%)	55 (8.0%)	69 (15.9%)	211 (9.1%)
Implant [*N*]	64 (5.3%)	24 (3.5%)	47 (10.8%)	135 (5.8%)
Others [*N*]	170 (14.2%)	117 (17.1%)	92 (21.1%)	379 (16.4%)

**Table 2 antioxidants-11-02424-t002:** Detailed information on primers (designed with primer blast) and the corresponding qPCR conditions. Primers were synthesized by Eurofins Genomics (Ebersberg, Germany).

Target	GenID	Primer	Efficiency	Amplicon	T_A_
*EF1α*	NM_001402.5	For-CCCCGACACAGTAGCATTTG	1.90	98 bp	56 °C
		Rev-TGACTTTCCATCCCTTGAACC			
*RPL13a*	NM_012423.3	For-AAGTACCAGGCAGTGACAG	2.24	100 bp	56 °C
		Rev-CCTGTTTCCGTAGCCTCATG			
*PADI4*	NM_012387.2	For-AGAGGTGACCCTGACGATGA	2.09	310 bp	56 °C
		Rev-CAGGTCTTCGCTGTCAAGCA			

## Data Availability

The data presented in this study are available in the article.

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
