# Peer review of "Nicotine and Cotinine Induce Neutrophil Extracellular Trap Formation—Potential Risk for Impaired Wound Healing in Smokers"

_antioxidants, 2022, doi:10.3390/antiox11122424_

Round 1
Reviewer 1 Report
This is an excellent paper that evaluates the role of nicotine and cotinine on neutrophil extracellular trap formation with a focus on potential risk for impaired wound healing in smokers
I have no objection on this paper and wish to congratulate on with the Authros for the excellence of their work
Author Response
We would like to thank the reviewer for his/her positive estimate of our manuscript. As suggested by the second reviewer and editor, the order of the manuscript was modified (M&M section was moved before the results section and some parts from the introduction were moved to the discussion). Overall the manuscript was double-checked for type-O‘s and doubled introduction of abbreviations.
Reviewer 2 Report
In my opinion Introduction section is too long and includes elements more associated with discussion section as well as the result section is confused with discussion section.
Authors sould describe with more details materials&methods section. Reagents and kits should have Cat. No.
Patients should be clearly describe according to exposure to tobacco smoke and diseases.
Was in the study group of patients who receive nicotine replacement? Because in case of the lack of this group the conclusions are too far.
The abbreviations are introduced many times.
Because the present study is retrospective, how authors collect the blood?
In my opinion verses 207-108 are more related to method section not to results section
Finally how many patients included into the study have diabetes? 90 or 84?
All antioxidants were determined in the mixture of CSE, nicotine and cotinine?
In my opinion whole manuscript should be rewritten and the material&methods section should be decribed with more details.
Round 2
Reviewer 2 Report
In current version I accept the paper